# New Sensing Technologies for Grain Moisture

**Omar Flor [1], Héctor Palacios [2,3,*], Franyelit Suárez [1] , Katherine Salazar [2], Luis Reyes [4], Mario González [5] and Karina Jiménez [6]**

1 Ingeniería Industrial, Facultad de Ingeniería y Ciencias Aplicadas, Universidad de las Américas, Quito 170125, Ecuador; omar.flor@udla.edu.ec (O.F.); franyelit.suarez@udla.edu.ec (F.S.)
2 Ingeniería Agroindustrial, Facultad de Ingeniería y Ciencias Aplicadas, Universidad de las Américas, Quito 170125, Ecuador; katherine.salazar.alvarez@udla.edu.ec
3 Ingeniería de Alimentos, Facultad de Ingeniería Mecánica y Ciencias de la Producción, Escuela Superior Politécnica del Litoral, ESPOL, Guayaquil 090150, Ecuador
4 Maestría en Agroindustrias, Universidad de las Américas, Quito 170125, Ecuador; luis.reyes@udla.edu.ec
5 SI2Lab, Universidad de las Américas, Quito 170125, Ecuador; mario.gonzalez.rodriguez@udla.edu.ec
6 Departamento de Investigación, Universidad de las Américas, Quito 170125, Ecuador; karina.jimenez@udla.edu.ec
* Correspondence: hector.palacios@udla.edu.ec or hapalaci@espol.edu.ec

**Abstract:** In this review, we present a description of conventional technologies and new advances for the estimation and sense of moisture content in grains. The operating principles, accuracies and response times are described. The review considers an exhaustive search of scientific developments and patent registrations. It was concluded that most of the new developments correspond to methods of which the measurement principles are based on the analysis of the electrical characteristics of the grains. In addition, new methods of image analysis have been implemented that provide measurements with reduced response times and with precisions of utility for its application in the agro-industrial field. In addition to this, wireless communication technologies have been implemented that allow the implementation of moisture measurement methods in moving grains within processing chains.

**Keywords:** moisture; grains; sensor; technologies

## 1. Introduction

Some varieties of grains have hygroscopic characteristics and are influenced by the relative humidity of their environment [1]. The humidity of the grains is directly associated with their quality, safety and durability. These parameters are important in the production, storage, processing and marketing of grains [2]. Temperatures and humidity outside the recommended ranges will allow for increased growth of microbes and insects in stored grains that affect the quality of grains and cereals. This aspect is relevant for all industrial processes that work with this type of product.

As shown in Figure 1, the percentage of humidity and temperature define specific regions. For example, when the temperature is above 18 °C and the moisture content exceeds a value of 14% this creates an ideal environment for the proliferation of insects, while temperatures below 18 °C and with humidity above 15% increase the growth of pests, insects and fungi [3], which causes an increase in the temperature of the grain mass and causes germination problems. The safety zone, according to Figure 1, is at humidity below 14% and temperatures below 18 °C. High levels of grain moisture in the harvest can lead to heat, fermentation, spoilage and a low flowering rate. To ensure safe storage, the grain should be dried as soon as possible after harvest and kept at lower moisture levels [4].

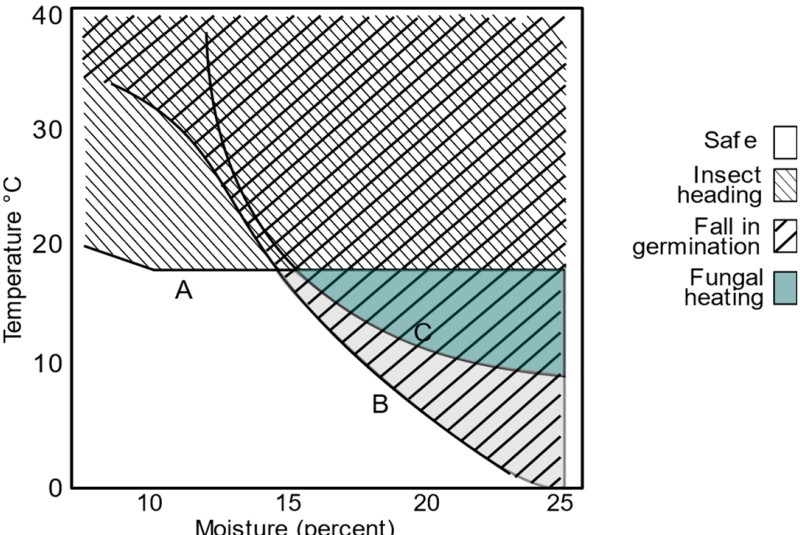

**Figure 1.** Effects in storage at different temperatures and moisture content.

Grain moisture content (MC) is one of the most important factors describing the biological activity of grains. Dry grains have a low respiration rate, while grains with higher moisture content tend to generate a suitable environment for microbial processes that result in higher $CO_2$ production rates. The moisture content of the grain in which the grains begin to breathe most intensely is called critical moisture (14.5–15% MC) [5]. When the moisture content of the grain exceeds a critical value, dew forms on the grain, initiating hydrolysis, respiration and enzymatic activities [6].

Due to physical, chemical and biological environmental factors, grains or other raw foods can begin to deteriorate immediately after harvest [6,7]. Post-harvest grain is a good medium for the growth of various microorganisms. The composition of microbial species in the grain is very diverse, but bacteria and fungi have the greatest impact on the grain quality and the storage period. Fungi begin to actively develop when the moisture in the grain is 15% to 18.5%. Bacteria begin to develop when the moisture of the grain exceeds 20% [8].

Since moisture is directly related to the biological behavior of grains, the sensitive and accurate determination of moisture content both in storage and at other stages of the production process is of great importance.

For the measurement of moisture in grains, there are direct methods that are very accurate (Figure 1), but they are also less accessible and require specialized facilities and equipment. Indirect measuring devices offer a wide range of solutions, are more accessible and less accurate [9]. Indirect methods have a fast response time and should be used taking into account the margin of error and test conditions for which they should be calibrated periodically.

Methods such as gravimetric, Karl Fischer titration, infrared radiation and others [10] are the most used and required by the specific legislations of each region, and are usually established in international standards such as ISO, USDA, etc. The methods described above offer convenient values of accuracy, repeatability and reliability.

Agro-industrial companies, producers and buyers use portable rapid measuring equipment; including methods such as capacitance, impedance, infrared spectrum and others. These devices have been developed for rapid detection and monitoring on production lines [11] and should be used with caution and knowledge of their errors, as well as being subject to calibration by certified bodies.

## 2. Methods

This section describes the direct and indirect methods for determining moisture content in grains shown in Figure 2.

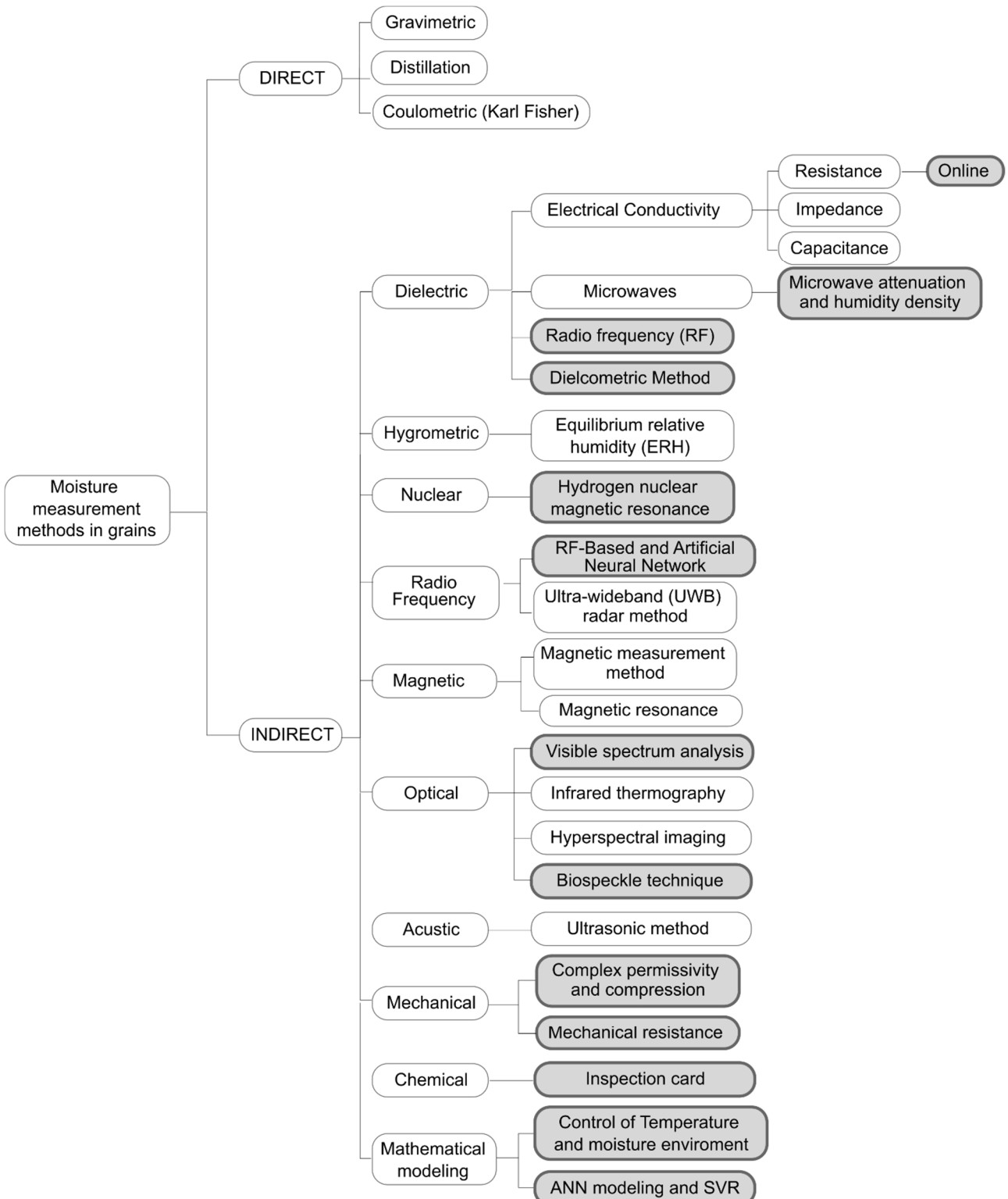

Advances in moisture determination methods and techniques in the last 15 years

**Figure 2.** Classification of grain moisture measurement methods.

Table 1 describes the direct methods and its operating principles; reference sources are indicated for each of these methods.

**Table 1.** Direct methods for moisture determination in grains and their principles of operation.

| Method | Operating Principle |
|---|---|
| Gravimetry/Oven [12,13] | Loss in mass by drying |
| Distillation [14,15] | At boiling temperature higher than water |
| Coulometry Method (Karl Fischer) [16,17] | Selective chemical reaction with water |

### 2.1. Direct Methods for Grain Moisture Measurement

Figure 2 presents the direct methods according to [18], characterized by measuring the change in weight by removing moisture from the grain for which these techniques use furnaces, heat sources and chemicals to remove the water of the species [19]. The operating principles for these methods are described in Table 1.

### 2.1.1. Gravimetric/Oven Method

The gravimetry/oven method, an international reference method for the determination of humidity, consists of drying a certain amount of 8 to 35 g of grains with at least 3 repetitions in a metal box with a lid that is placed in a hot air oven (controlled oven temperature) at a temperature of 130 °C for $38 \pm 2$ h. After the drying time, it is left in the oven for 30 to 45 min to cool, then the weighing (Equation (1)) of the dry grain sample is carried out [12].

$$\%H_{bh} = \frac{m_h - m_s}{m_h} \cdot 100 \tag{1}$$

where $H_{bh}$ is the wet basis content (MC), $m_h$ is the mass of the wet sample (g) and $m_s$ is the mass of the dry sample (g).

The advantages of this method are: high reliability with uncertainty levels around 0.4 $\%H_{bh}$ [13], an economic process, traceability to the international system (mas) and low measurement uncertainty. As disadvantages this method destroys the sample, the evaporation of the volatile material is evident and the execution time is long.

### 2.1.2. Distillation Method

A portion of grains, grasses or ground or whole cereals are immersed in oil and this sample is exposed to heat in which the boiling temperature is higher than that of water. The water vapor in the sample condenses and is measured in a graduated cylinder. The water content measured in the graduated specimen corresponds to the moisture content expressed as a percentage [14]. The weight loss of the sample can also be evaluated [15]. An outline of this method is shown in Figure 3.

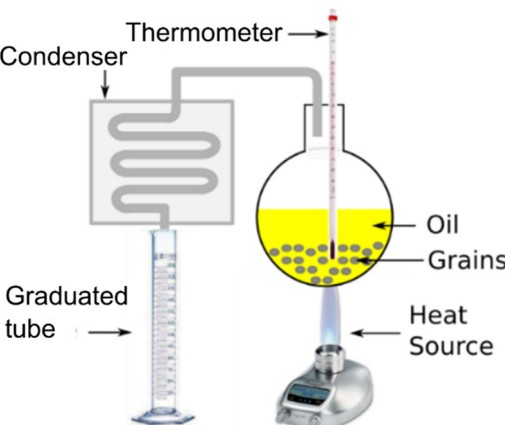

**Figure 3.** Diagram of the method of determination of humidity by distillation.

### 2.1.3. Coulometry Method (Karl Fischer KF)

It is used as the main reference method for quantifying water content in multiple products, including cereals [16]. Its fundamental principle is based on the Bunsen reaction with iodine and sulfur dioxide ($SO_2$), then the percentage of moisture is calculated using the reference formulations of this method [17].

Karl Fischer's method (KF) is based on a chemical reaction that consumes water. This method requires the test portion to be weighed and added directly to the titration cell. This is the preferred method for its reliability and minimal pollution caused by atmospheric moisture. The chemistry of KF reactions has been documented and is presented in Equations (2) and (3) [16]

$$CH_3OH + SO_2 + RN = [RNH]SO_3CH_3 \tag{2}$$

$$H_2O + I_2 + [RNH]SO_3CH_3 + 2RN = [RNH]SO_4CH_3 + 2[RNH]I \tag{3}$$

where $RN$ is the imidazole base, $CH_3OH$ is methanol, $SO_2$ is sulfur dioxide, $SO_3CH_3$ is naphthalenedisulfonic acid dimethyl ester, $SO_4CH_3$ is dimethyl sulfate, $H_2O$ is water, $RNH$ using this chemical equation balancer.

Sulfur dioxide reacts with alcohol to form an ester, which is neutralized with a base. The anion of alkyl sulfuric acid is the reactive component and is present in the reagent of the KF method. The titration of water constitutes the oxidation of alkyl sulfate by iodine. This reaction consumes water [16]. Therefore, two important prerequisites must be met to ensure a stoichiometric course of the KF reaction. The first requirement is the presence of an alcohol suitable for fully esterifying sulfur dioxide. Methanol is the preferred choice, because the KF reaction is stoichiometric and fast. In one of their methods, methanol is mixed with formamide, which dissolves the sugar and helps distribute the protein. The formamide content should not exceed 50% to maintain reaction conditions [20].

### 2.2. Indirect Methods for the Determination of Moisture Content in Grain

Indirect methods (Table 2) have been widely used thanks to their practicality and the speed with which results are obtained (a few seconds). Most grain and cereal production and commercial sectors have such equipment; however, they should be used with caution and caution be paid to their error ranges [14].

**Table 2.** Indirect methods for determining moisture in grains and their principles of operation.

| Methods | Operating Principle |
|---|---|
| Electrical resistance/impedance [21,22] | Change in electrical resistance/impedance |
| Dielectric constant change [23,24] | Dielectric behavior of a grain sample in a capacitor |
| Microwave [25,26] | Measurement of microwave free space, its attenuation and phase determination |
| Magnetic measurement method [27,28] | Is based on the orbital motion of the electrons due to the presence of an applied external magnetic field |
| Magnetic resonance [29–31] | Analysis of free-induction decay (FID) and, in some cases, spin echo (SE) outputs |
| Ultra-wideband (UWB) radar method [32,33] | Reflection coefficient of the sensor and the complex permittivity of the grain are analyzed |
| Ultrasonic (Acoustic Method) [34] | Analysis and comparison of the incident sound wave in grains at frequencies from 5 to 50 kHz |

**Table 2.** *Cont.*

| Methods | Operating Principle |
|---|---|
| Infrared thermography (IR) [35] | Detect difference in IR emitted by a dry sample and a wet sample |
| Infrared spectroscopy [32,36,37] | Analysis of radiation absorption in the infrared spectrum |
| Hyperspectral imaging (HSI) [38,39] | Evaluates drying kinetics of the drying method |
| Equilibrium relative humidity (ERH) [40–42] | Determines the humidity as a function of the ratio between the EMC and ERH value at fixed temperature |

2.2.1. Electrical Resistance/Impedance Method

This method essentially consists of measuring the resistance to the flow of an electric current through the grain that is located between two metal electrodes. The electrical resistance decreases rapidly as the moisture content in the grain increases. This method, therefore, generates a scale and a considerable variation of the resistance that is used to determine the moisture content of the grain.

Figure 4 shows the curve describing the relationship for a wheat case where resistance is measured in megaohms [21].

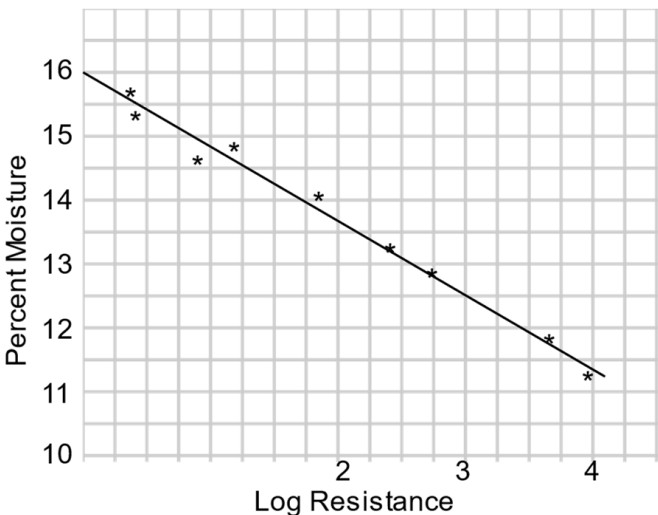

**Figure 4.** Relationship between moisture content and electrical resistance for wheat measured at 75 °F.

There are studies such as [22] in which parallel plates are used and the impedance is analyzed, which is the equivalent of the resistance when using alternating current. This impedance has real and imaginary components and they use a semi-empirical equation that takes into account the impedance and the phase angle to estimate the moisture content (CM) of yellow corn. Devices operating under this principle are characterized by being fast, do not destroy the sample and are widely used in analytical laboratories for industry and agriculture [12].

2.2.2. Dielectric Constant Change

Determining the percentage moisture content with this method requires a capacitor that is composed of two opposing charges consisting of conductive plates separated by an insulator called a dielectric [23]. The region or space between the plates is influenced by an electric field that interacts with the grain sample offering a dielectric behavior. Figure 5 schematically shows a simple parallel plate capacitor, with an AC power supply.

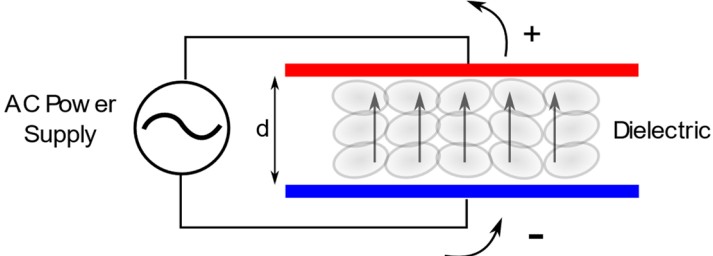

**Figure 5.** Grains in single plate capacitor.

A capacitor is a device that can hold or store an electrical charge. The presence of the grains influences the behavior of the single-plate capacitor (Figure 5). Equation (4) describes the ratio of capacitance [43].

$$C = \frac{KA}{4\pi d} \tag{4}$$

where $C$ is capacitance (F), $K$ is dielectric constant, $A$ is the area (m²) and $d$ corresponds to the distance between the plates (m).

The higher the dielectric capacity of the grain, the higher the dielectric constant $K$, and therefore more energy from the electrical network is required to charge the capacitor plates and maintain the behavior of the capacitor. This change in the dielectric constant is associated with the moisture content in the grain, allowing for high speed, high reliability, economy, portability, easy maintenance and in-line measurement [24].

### 2.2.3. Microwave Method

This method is based on microwave free space measurements that involve the determination of attenuation and phase change, for which functions independent of the grain permittivity density are also presented and that allow a reliable detection of the humidity applicable to the moving grain in which a variation of the bulk density is evidenced [25,26]. The dielectric behavior $\varepsilon$ of the grain is determined by considering Equation (5).

$$\varepsilon = \varepsilon' - j\varepsilon'' \tag{5}$$

where $\varepsilon'$ is the dielectric constant and $\varepsilon''$ is the dielectric loss factor.

The permittivity in this method refers to the relative complex permittivity. This property of grains and cereals is not only a function of the moisture content $M$ but also of the frequency f of application of the electric field, the temperature $T$ of the grain and its bulk density $\rho$. Therefore, a plane wave passing through a grain layer thick $d$ will interact with the granular material as shown in Figure 6, where $E_i$ represents the electric field of the incident wave, $E_r$ is the electric field of the reflected wave, $E_t$ is the electric field of the transmitted wave, $\Gamma$ is the reflection coefficient, $\tau$ is the transmission coefficient and d is the distance the wave must travel [44].

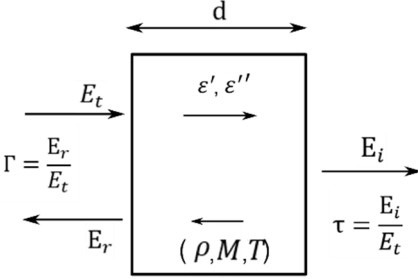

**Figure 6.** Diagram of wave-dielectric material interaction.

### 2.2.4. Magnetic Measurement Method

A well-known magnetic property of water is diamagnetism, based on the orbital motion of electrons due to the presence of an applied external magnetic field. Diamagnetism has very weak properties and is very difficult to detect, because the magnetic susceptibility of water is of a very low value. A phenomenon has been observed whereby a secondary magnetic field was generated from a large volume of water when an external main magnetic field was imposed on the water, which was proportional to an increase in frequency above 400 Hz [27]. The secondary magnetic field of water was in opposition to the main magnetic field with a phase change of 180 and was not affected by conductivity [28]. Using the property described above for low-frequency water, this method has been developed to determine the water content in grains.

The approached method requires for its measurement a magnetic coil and a magnetic resistance sensor (Figure 7). When grain is exposed to alternating low-frequency magnetic fields, a secondary magnetic field generated by the grain is observed. The equipment in Figure 7 has specific dimensions described in [19]. Compared to a sample of pure water, the secondary magnetic field was obtained by vector analysis. This analysis is used as a reference method for determining the percentage of moisture in grains.

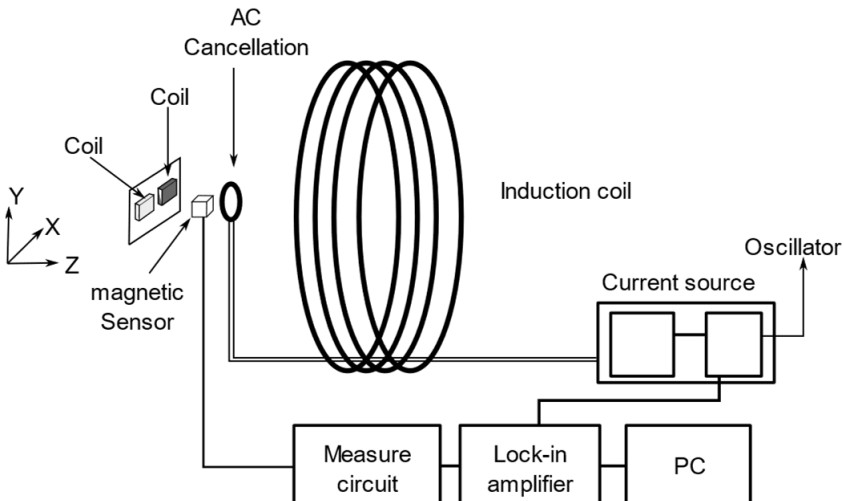

**Figure 7.** Schematic diagram of a magnetic measurement system that detects the moisture content of the grain.

### 2.2.5. Magnetic Resonance Method

This method uses magnetic resonance imaging (MRI) of the hydrogen atom in the grain and provides accurate and sensitive measurement. From the resonance signal, its intensity is measured in relation to the total number of protons in the grain [29]. However, MRI is expensive and inadequate for routine monitoring.

MR technology is based on the analysis of the magnetic moments of atomic nuclei. The nucleus of an atom is made up of a certain number of protons and neutrons (the only exception is the hydrogen nucleus, which is a single proton) and has a complex collective rotational motion around its own axis, as shown in Figure 8. This rotation gives the proton, from the mechanical point of view, a kinetic moment called spin represented by the vector S, with the same direction as its axis of rotation and magnitude expressed in (Hz), and, from the electrical point of view, a magnetic moment represented by the magnetization vector $\mu$, the magnitude of which is expressed in Tesla (T) [30].

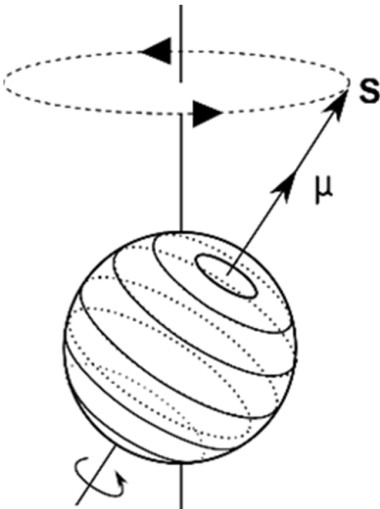

**Figure 8.** Moment resultants on a proton.

The basic components required for the measurement of moisture content are shown in Figure 9 [31]. The system includes: a system that produces a magnetic field, an RF transmitter that supplies a signal according to the selected cores in the sample and according to the external magnetic field (Figure 9), a receiver that detects the signal generated by the precession motion of the magnetization of the spins in the sample and sends the FID signal to a processor in a system also composed of a sequencer (pulse programmer), a data acquisition system and a signal processor and a display [28].

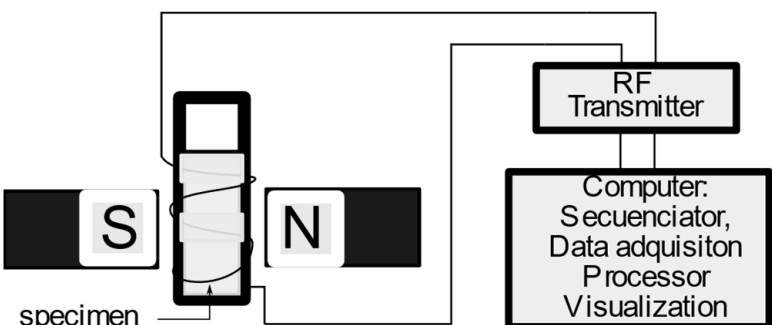

**Figure 9.** Schematic diagram of basic equipment for measuring RM.

2.2.6. Ultra-Wideband (UWB) Radar Method

This method allows a broadband electrical signal to be emitted in a cylindrical vessel and the reflection coefficient of the sensor and the complex permittivity of the grain are studied using formulations [32]. The sensor reflection coefficient was measured using an ultra-wideband radar module (UWB) and the moisture content of the grains was calculated from the complex permittivity using a density-independent model [33].

To measure the wideband coefficient a UWB radar module is used; schematic diagram shown in Figure 10. A circulator alternates the signal from 3.1 to 4.8 GHz emitted to the sensor (1–2) through a transmission line through which the signal passes, and (2–3) is also returned to the UWB radar module. The shape of the sensor allows it to be inserted into an immobile volume to be analyzed [45].

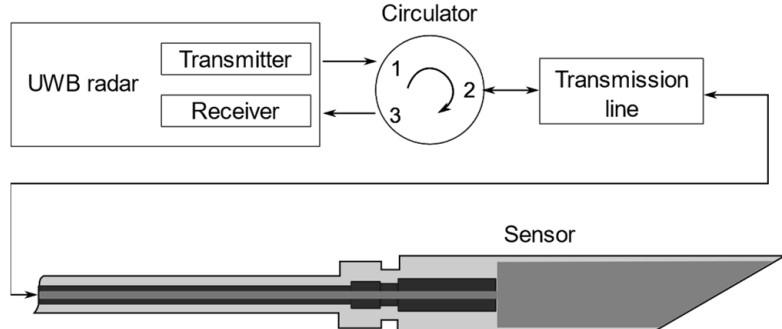

**Figure 10.** Schematic of portable UWB detection system.

For the determination of the moisture content in grains, a density-independent calibration function has been adopted. The calibration function is a general method of determining the moisture content of the grain from the complex coefficient, which can eliminate the influence of bulk density. In this method, Equation (6) is used.

$$\psi = \sqrt{\frac{\varepsilon''}{\varepsilon'(a_F\varepsilon' - \varepsilon'')}} \tag{6}$$

where $\psi$ is the grain density-independent calibration function, $a_F$ is a frequency factor, $\varepsilon\prime$ is the dielectric constant and $\varepsilon''$ is the loss factor.

### 2.2.7. Ultrasonic Method (Acoustic Method)

The technique used is based on the propagation of acoustic waves through the porous material that are emitted by an ultrasonic transducer system. The acoustic signal propagates through an emitting transducer, conveniently coupled to the material, with an excitation voltage of 20 Vrms; the magnitude of the transmitted signal excites the transducer receiver, delivering a voltage response that correlates with the moisture content (Figure 11) [34].

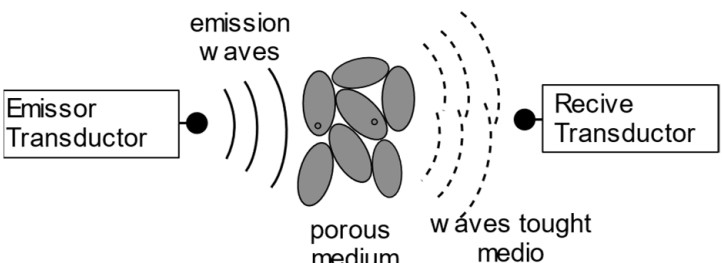

**Figure 11.** Schematic of the transducer arrangements facing each other within the method.

The measurement of the amplitude of the sound wave signal is expressed in "sound pressure level" (SPL) measured in decibels (dB) and using Equation (7) [46].

$$SPL = 10log_{10}\left(\frac{p}{p_{ref}}\right)^2 \tag{7}$$

where $p$ is the effective pressure (N/m$^2$) and $p_{ref}$ is the reference sound pressure (commonly $2 \times 10^{-5}$ N/m$^2$). There is the possibility to perform the analysis with two sound sources for which the concept of total effective sound pressure is used as discussed [46].

This measuring principle is a valuable tool, because it can also indicate differences in moisture content when scanning different cross-sections. It does not provide any information on the distribution of moisture as a function of depth and only gives an average value. In practice, its application is limited, since the translation of the transferred signal into moisture content requires a situation-specific calibration [47].

2.2.8. Infrared Thermography (IR) Method

This method uses information from radiation emitted in the thermal infrared (IRR) range (Figure 12) of the electromagnetic (EM) spectrum [35]. This information is converted into temperature by a computer [48], as shown in Figure 13. In general, all elements of the landscape such as vegetation, soil, water and people emit TIR radiation in the 3.0 to 14 μm portion of the EM spectrum [49].

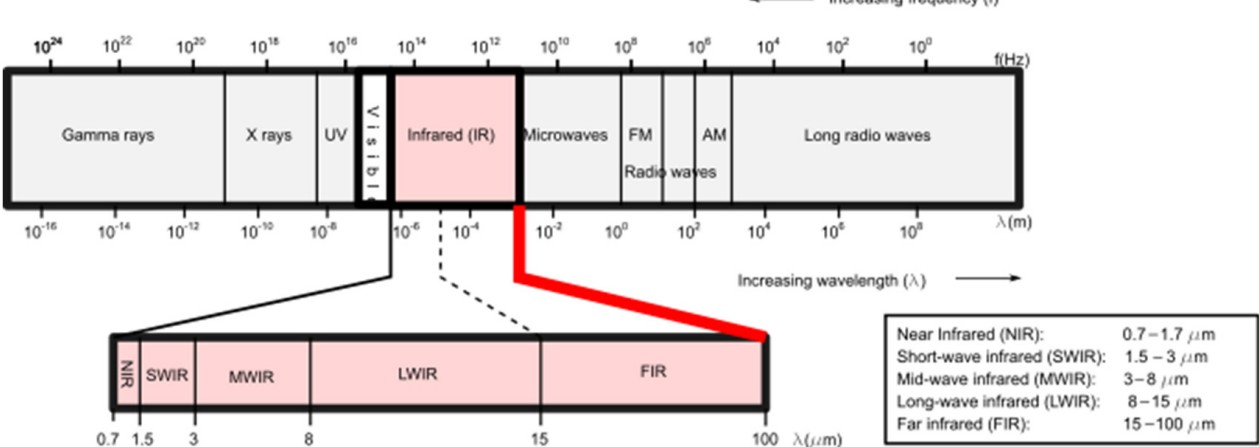

**Figure 12.** The electromagnetic (EM) spectrum. As evidence, the infrared (IR) region in which the reflected IR (0.7–3.0 μm) and emitted IR (3.0–100 μm) are more detailed.

The measurement principle is based on detecting the difference in the IR emitted by a dry and a wet sample. An infrared camera detects thermal energy (mainly in the range of 3 to 20 μm) emitted by the surface of an object and transforms it into a visible "temperature map" of the surface.

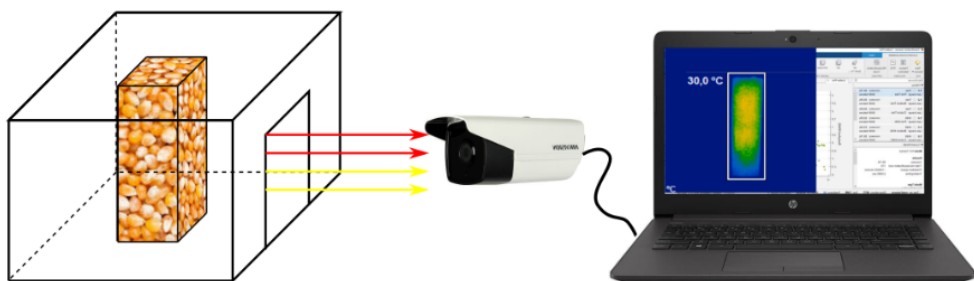

**Figure 13.** System employed to capture the thermographic images of grain samples.

The amount of radiation emitted depends on the characteristics of the surface and the temperature of the object. As water has a higher specific heat capacity than most materials, if a wet sample is heated or cooled it will take longer to return to equilibrium with the environment than a dry sample [50–52].

### 2.2.9. Infrared Spectroscopy

This technique is based on the analysis of the near infrared (NIR) spectrum, it allows the simultaneous analysis of the micro components of the product. This rapid and non-destructive technique determines the main components from the radiation spectra emitted by the product [32]. Specific organic molecules absorb wavelengths of near-infrared light energy. Absorptions are directly related to the concentration of organic molecules in the sample (Figure 14). The above relationship correlates with a primary technique, also known as wet or bench chemistry, to achieve the linear relationship between molecular absorptions and actual concentrations of constituents [36]. NIR spectroscopy is generally accepted as a technique for industrial applications and has been successfully used for rapid analysis of moisture, protein and fat content in agricultural and food products [37].

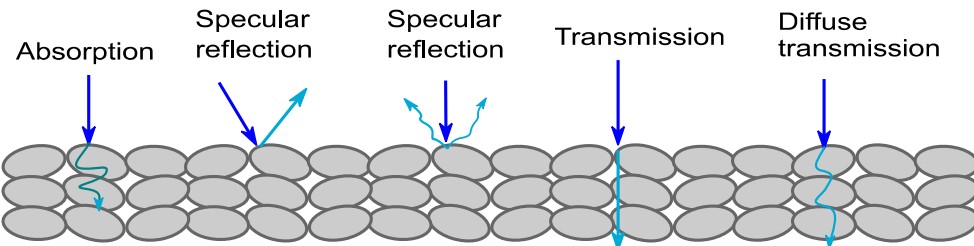

**Figure 14.** Interaction of light with solid bodies.

### 2.2.10. Hyperspectral Imaging (HSI)

Hyperspectral imaging is a useful method for non-destructive measurement and visualization of the moisture content in food materials. It provides a basis for evaluating the drying kinetics of the drying method. HSI is produced in imaging spectrometers, commonly known as spectral imaging or spectral analysis, that combine conventional imaging and spectroscopy [39].

An advantage of HSI is its ability to perform non-destructive measurements of irregularly shaped objects during the "in-line" drying process. The hyperspectral reflectance imaging system consists mainly of a hyperspectral imaging unit, a light source, a sample manipulation platform and a computer to control the camera and acquire the images. (Figure 15) [53].

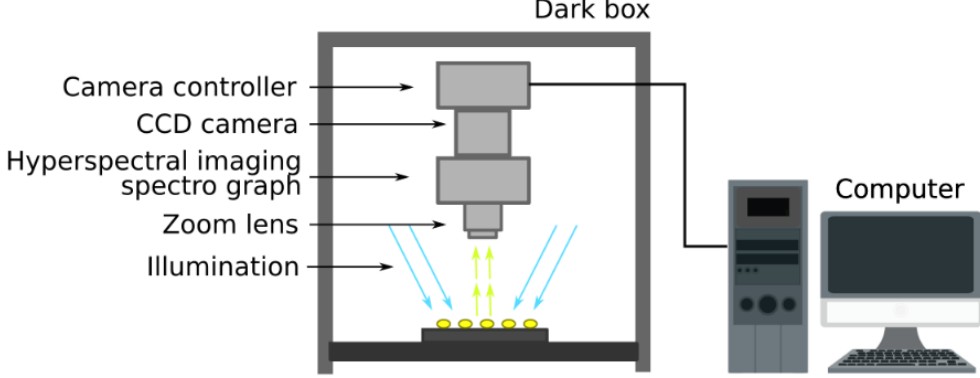

**Figure 15.** Schematic of the hyperspectral reflectance imaging system.

### 2.2.11. Equilibrium Relative Humidity (ERH)

Relative humidity (RH) sensors are reasonably easy to use as a means of estimating the water activity of food products. Sensors are usually based on capacitance or resistance (Figure 16) and operate over a wide enough temperature range (e.g., 0–50 °C) that is suitable for most applications [40]. Its measurement range is typically approximately 20% to 90% RH, with an accuracy of ±2% to 5% in the midrange, with potentially greater variability

at the end of the HR range. Temperature can be routinely measured with an accuracy of ±1.0–1.5 °C. Due to the high level of dependence of vapor pressure on temperature, this is an important consideration when selecting an HR meter [41].

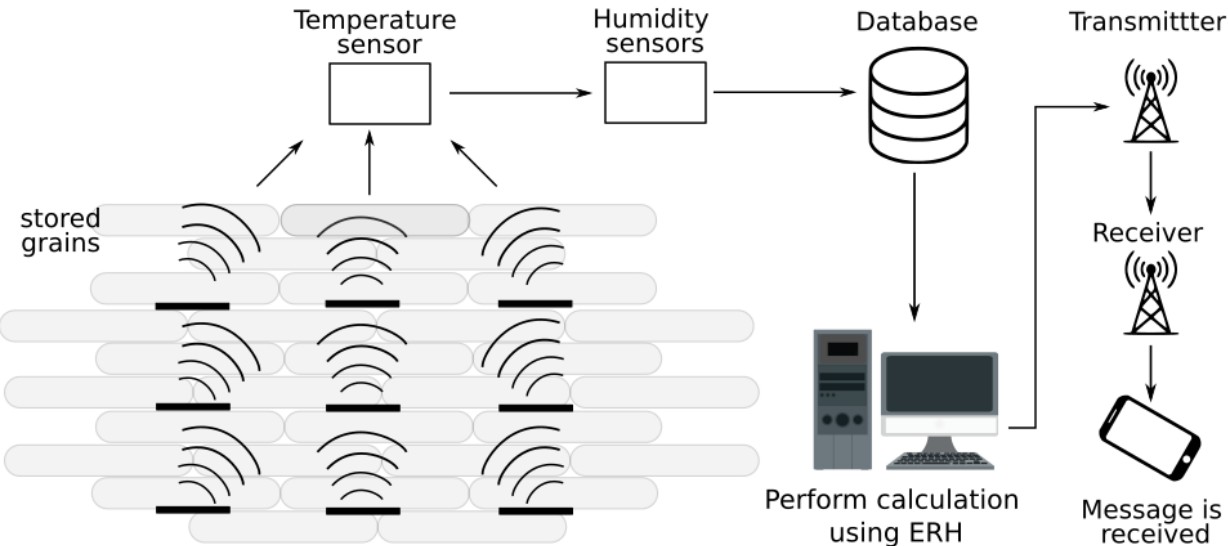

**Figure 16.** Schematic of system to sense moisture grain with ERH calculus.

The ERH equation is used to find the relative moisture content and determine the moisture content by interacting with 3 or 4 layers of grains, showing successful results for the case of evaluation with various types of rice [38].

The Oswin equation was used to express the relationship between the EMC value and ERH at a fixed temperature (Equation (8)).

$$M = A\left(\frac{ERH}{1-ERH}\right)^B \tag{8}$$

where $M$ is the moisture content in %(MC), $ERH$ is the equilibrium relative humidity in decimal and $A$ and $B$ are constants. The relationship between parameter $A$ and $B$ and temperature was further analyzed by regression analysis [54].

### 3. New Techniques for Determining Moisture in Grains

Some advances have been developed in methods based on the dielectric behavior of grains [55] and several developments have been made to measure moisture content using low-cost wireless technology operating in the 2.4 GHz and 915 MHz ISM bands such as the wireless sensor network (WSN) and radio frequency identification (RFID) [56].

New techniques including machine learning and the use of neural networks have been employed along with dielectric methods to improve the prediction and estimation of grain moisture [57]. Advances have also been found in the determination of visible spectrum imaging and grain surface characteristics [58,59].

Table 3 shows a description of the methods and their respective principles of operation. In some cases, where indirect methods are named, implementations and advances are explained.

**Table 3.** Methods for moisture determination in grains using Information and Communications Technologies ICT.

| Method | Operating Principle |
| --- | --- |
| Visible spectrum analysis [58] | Image analysis of the statistics of intensity distribution in images compared with established humidity patterns |
| RF-based moisture content and Artificial Neural Network [60,61] | RF signal strength analysis and use of Random Forest method with one input feature (RSSI/WSN) |
| Prediction using ANN and SVR modeling techniques [57,62] | Estimation by neural network training with inputs: number of days after sowing, air temperature, relative air humidity, hourly wind speed and 6 h precipitation |
| Grain moisture determination by complex permittivity and compression [63] | Electrical analysis of complex permittivity in compressed grain sample |
| Microwave attenuation at 10.5 GHz and humidity density [64] | Uses microwave attenuation at 10.5 GHz and humidity density |
| Temperature and humidity control [65] | The WU equation is used to relate grain moisture to relative humidity and temperature |
| Online measurement by resistance sensor [66,67] | Single-grain analysis with reliable and accurate measurements from an adaptation of resistance method, suitable for measuring on production lines |
| Indirect determination of moisture using the biospeckle technique [68] | Use of laser light, image analysis and moisture value expressed as modified correlated matrix and Moment of Inertia (MOI) |
| Moisture content in the mechanical properties of grains [69–71] | Determination of moisture in grains as a function of the variation of mechanical properties in compression |
| Moisture content measurement with inspection card [72] | Indicator cards are used, which discolor on contact with the moisture in the grains, the colour is compared and a moisture reference is obtained |
| Evaluation of grain moisture content by the dielcometric method [73,74] | Use of mathematical modeling and graphical representation of electrophysical processes under the effect of an electric field |

### 3.1. Visible Spectrum Analysis

Based on the analysis of images of grain samples that have been obtained with a controlled light environment, the moisture content has been correlated with morphological characteristics such as dimensions, color and weight.

Images are acquired using the system components shown in Figure 17 [58]. The images are obtained inside a closed box (1), avoiding the influence of external light in a specialized container considering only healthy grains. The sample is inserted under a scanner (2) and an electronic scale under the sample. The sample is placed in the detection area of the scanner (3) and the data and segmented images with their characteristics obtained from the sample images are recorded with the help of a computer (4).

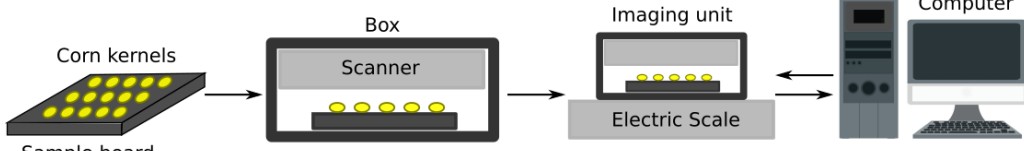

**Figure 17.** System operating procedure. 1–2, sampling; 3, imaging and weighing; 4, analyze.

In addition to morphological characteristics such as area and perimeter, the maximum, minimum, mean and standard deviation of RGB primary colors (red, green and blue) and derived HSI color values (hue, saturation and intensity) are also included. The moisture content is obtained from the analysis of correlations between moisture content, morphological characteristics and geometric characteristics, showing a linear trend that increases with increasing moisture content.

### 3.2. RF-Based Moisture Content and Artificial Neural Network

This method uses 4 GHz frequency radio frequency (RF) transceivers, ZigBee communication standard and an 868 to 915 MHz UHF RFID transceiver used for the classification and prediction of moisture content in grains with the use of artificial neural network (ANN) models [60]. The received signal strength (RSSI) is used to perform moisture content prediction, which has been successful in studies on rice varieties. The processed data is used as input for different ANN models [61] such as the support vector machine (SVM), K-Nearest Neighbor (KNN) [75], Random Forest [76] and multilayer perceptron (MLP) [77]. The results show that the Random Forest method with an input feature (RSSI/WSN) provides the highest accuracy of 87% compared to the other four models. A combination of one of the other models with the two input features (RSSI/WSN and RSSI/RFID) using RSSI with the wireless WSN sensor network and RFDI radio frequency identification provides an accuracy of more than 98% [56].

### 3.3. Prediction Using ANN and SVR Modeling Techniques

For the determination of moisture content in wheat, a new methodology has been developed that uses multilayer perceptron neural network (MLP) and support vector regression (SVR) techniques [62]. Five inputs are used to train the neural network: the number of days after planting, air temperature, relative humidity, wind speed per hour and 6 h precipitation. The results indicated that the MLP model outperformed the SVR model in determining the moisture content of wheat, concluding that the MLP model is useful for estimating wheat moisture content at harvest time [57]. The diagram in Figure 18 presents at the end of the network an output corresponding to the estimated percentage moisture content for the grain.

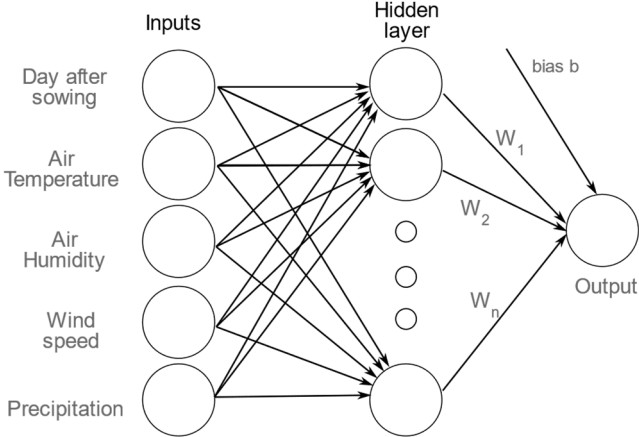

**Figure 18.** Basic neural network diagram for neural network development.

### 3.4. Grain Moisture Determination by Complex Permittivity and Compression

To improve the accuracy of grain moisture determination, the seed grain is compressed into a self-supporting bulk sample and its complex permittivity is determined by analysis of its indirect electrical character. The arrangement of the grains reduces empty spaces improving the homogeneity of the electrical behavior of the grain. Considering the real and imaginary parts of the complex permittivity, the moisture content of the grain is determined. The force that compresses the grains does not affect the complex permittivity, which improves the accuracy of the measurement of the moisture content of the grain [63].

An outline of the system used in this method is shown in Figure 19, showing the grains compressed by the reinforcing steel jacket and using two calibration planes.

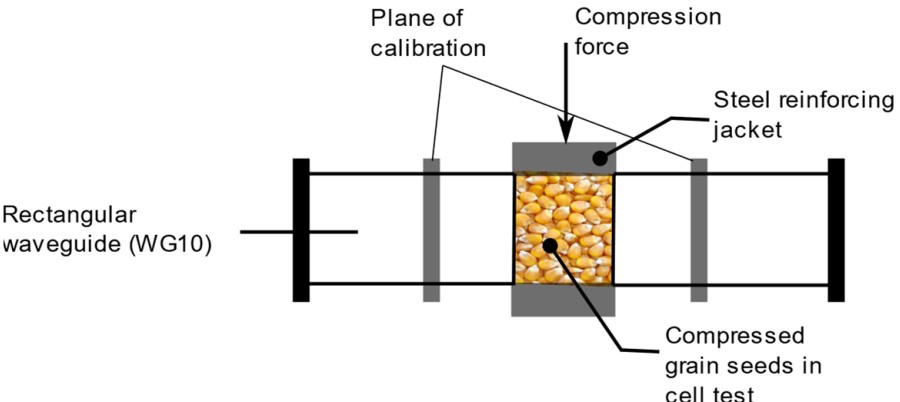

**Figure 19.** Waveguide cell setup for the compressed grain bulk sample.

### 3.5. Microwave Attenuation at 10.5 GHz and Humidity Density

Considering the dielectric behavior of grains of brown rice, brown rice and barley with moisture contents ranging from 11% to 27%, 11% to 18% and 11% to 21%. A prototype grain moisture meter has been developed using microwave attenuation at 10.5 GHz and moisture density. A third-order polynomial regression model was proposed to describe the relationship between dielectric properties and moisture density. The prototype grain moisture meter consisted of a 10.5 GHz dielectric resonator-type oscillator, a horn antenna, a rectangular sample holder, a load cell, a temperature sensor, a detector and a digital voltmeter. The calibration equation for measuring grain moisture content was developed and estimated with Korean short grain paddy rice (12% to 26%). The coefficient of determination, standard prediction error (SEP) and bias were 0.986, moisture content 0.52% and moisture content 0.07%, respectively [64].

### 3.6. Temperature and Humidity Control

This method allows the measurement of the moisture content of the grain from an internal point of the silo based on the control of the temperature and humidity in the grain stored in silos. Considering the basis of relative humidity equilibrium, a mathematical model has been established that relates the equilibrium water content of the grain from the inner point of the silo to the temperature and humidity.

In the 1980s, the mathematical equations for regulating humidity and heat in quantity of grains and expressed in relative humidity and absolute humidity were proposed, which gave the relative (absolute humidity) [65]. The numerical correlation between relative humidity, grain humidity and temperature has established the technical basis of mechanical ventilation; a development known as Wu's model and corresponding to Equation (9).

$$ERH_r = exp\left\{ \frac{\frac{D}{222}\left(e^{\frac{Bi-M}{Ai}} - e^{\frac{B2-M}{A2}}\right)\left(1737.1 - \frac{474242}{273+t}\right) + \left(1 - e^{\frac{Bi-M}{Ai}}\right) + 202}{87.72} \right\} \qquad (9)$$

where $ERH_r$ is the equilibrium relative humidity of the grain (%), $M$ is the moisture content of the grain (% wet basis), $t$ is the grain temperature (°C) and $Ai$, $Bi$, $B2$, $D$ are the parameters of Wu's model [65].

### 3.7. Online Measurement by Resistance Sensor

Using the indirect method of determining grain moisture by electrical resistance [66], an adaptation has been implemented for the analysis of a single grain, which has improved the reliability, capacity and accuracy of the measurement. In addition, this type of device has been adapted to the production lines to take measurements at regular intervals. A schematic of the device is shown in Figure 20, which uses two electrical resistors on rotating cylinders and also serves as a conveyor for the grains. The distance δ between the rollers is a fundamental parameter in the process.

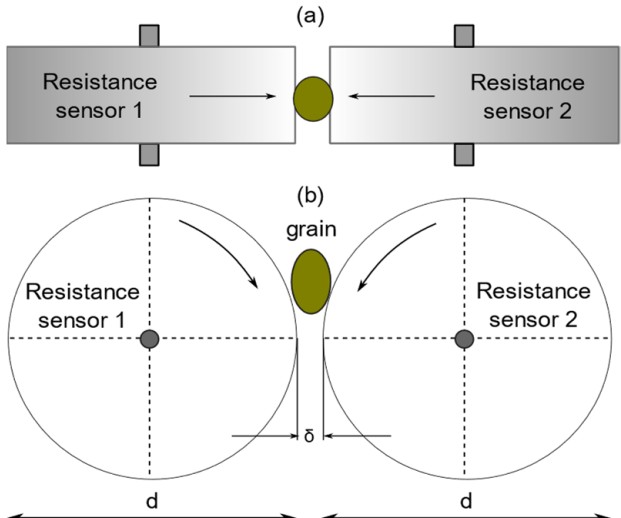

**Figure 20.** Structure of the single-grain moisture sensor: (**a**) top view; (**b**) front view.

The relationships between the moisture content and the resistance waveform were determined in real time based on a method of analytical calculation of the peak value and peak area of the waveform that correctly showed the electrical measurement properties of the grain. The results showed that the width of the space between the electrodes had a great influence on the performance of the sensor. In addition, an in-line measuring device was developed based on the experimental analysis and calculation method and the results of tests both in the laboratory and in the field for different grains showed that the absolute error of the real-time online measurement is within ±0.5% in moisture content variable in a humidity range of 10% to 35% on a wet basis and a temperature between −20 and 50 °C [67].

### 3.8. Indirect Determination of Moisture Using the Biospeckle Technique

Figure 21 shows an outline of the components needed to be analyzed with the biospeckle technique. The graph depicts a device that emits a laser light in such a way that when it hits a mirror the light bounces off the grains to be analyzed, the images of which are captured by a camera and then subjected to image processing and interpretation of the data [65–67].

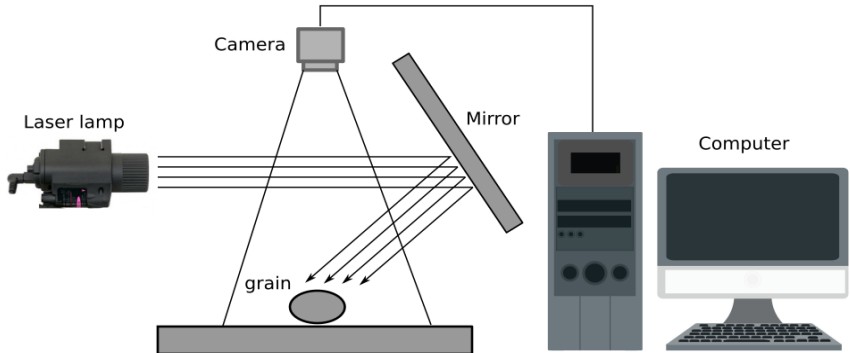

**Figure 21.** Schematic diagram of the biospeckle method of analysis.

### 3.9. Moisture Content in the Mechanical Properties of Grains

It has been proven that the effect of the moisture content influences the mechanical properties of grain structures [69–71]. The moisture content influences the values of maximum compressive strength, proportional deformity modulus and elastic coefficients considering the variety (Crambe abyssinica Hochst) under compression in a natural position and at rest; Crambe grains with moisture contents that had values from 0.1547 to 0.0482 decimal dB and dried at 40 °C. The samples were subjected to compression over a period of one hour. The samples were subjected to uniaxial compression between two parallel plates in a natural resting position. To control the amount of moisture determined, the gravimetric method was used to monitor the reduction of moisture content during drying (weight loss).

The compressive force required to deform the Crambe grains decreases as the moisture content increases, as shown in Figure 22. The proportional modulus of deformity increases as the deformation decreases, giving values between $(0.09–0.27) \times 10^2$ MPa. The sigmoidal model described by the Taylor series adequately represents the compressive strength of the Crambe grains in the natural resting position with a moisture content of less than 0.0813 dB [71].

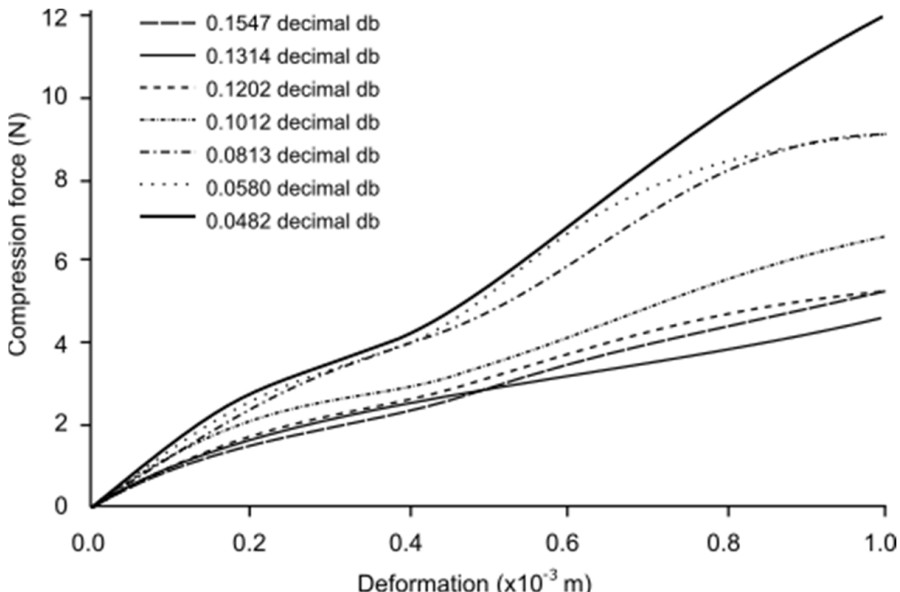

**Figure 22.** Strength as a function of deformation in Crambe (Crambe abyssinica) grains with different moisture contents.

Figure 22 shows the performance of the force needed to achieve a deformation equivalent to one micron. The graph shows that as the humidity increases, the grain deforms with the application of a lower force.

### 3.10. Moisture Content Measurement with Inspection Card

A method for measuring the moisture content of rice using a moisture inspection card was proposed and evaluated. Tests were performed for different duration and moisture content of rice, as evidenced by the discoloration of cobalt chloride coated paper. The indicator paper is soaked with a solution of cobalt (II) chloride and methyl alcohol and then dried at 60 °C. Experiments with rice and indicator paper were conducted in a flask with 25%, 50% or 75% headspace, with the lid closed for 30 or 45 min [72]. The results of the experiment to measure the change of coated paper in the rice paddy with a moisture content of 10–24% wb showed that the color of the paper slowly changed from blue to pink with an increase in moisture content (Figure 23).

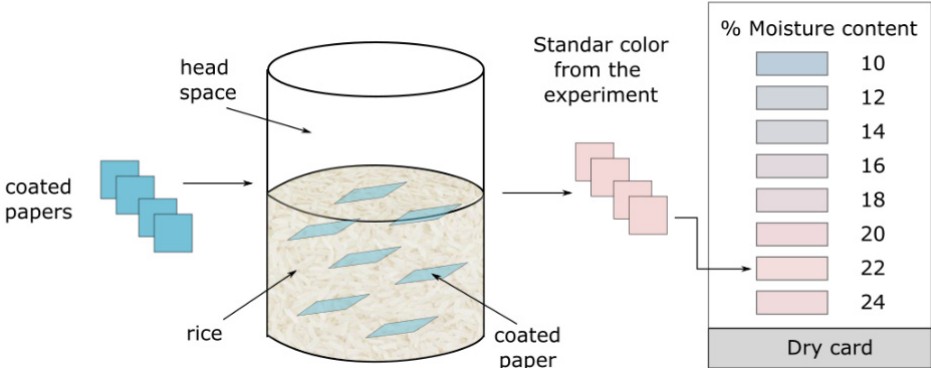

**Figure 23.** Moisture inspection card.

Card color comparisons in the CIE color system found that the color value of experiments with a headspace of 75% for 30 min was not significantly different from that used with a headspace of 50% or 25% for 45 min, although for the entire headspace levels the duration of the test of 45 min had more color than for 30 min with the different moisture contents of the rice. Therefore, using a 75% headspace for 45 min produced a significantly higher color value, so these conditions would be sufficient to allow the moisture content of the rice to balance with the relative humidity in the headspace. The L values of coated paper are converted into RGB values for printing color images to produce moisture inspection cards. This alternative can be replicated at low cost and could be reused, making it suitable for farmers in rural areas [72].

### 3.11. Evaluation of Grain Moisture Content by the Dielcometric Method

With an idealized mathematical model to monitor the moisture content of the grain, the dielcometric method graphically represents the electrophysical processes that have been studied in the weevils' grains under the effect of an electric field (Figure 24).

The relationship between the moisture content and the electrical parameters of the grain can be expressed analytically by equivalent circuits. An analysis of certain techniques and methods of effective use of the dielcometric method is presented. It is considered an example of the development of an electrical device to monitor the moisture content of grain with an instrumental error of less than 0.1% [73].

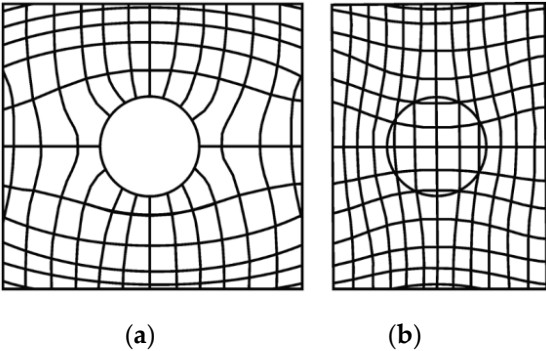

(**a**)                    (**b**)

**Figure 24.** Representation of forces and equipotential lines of uniform electric field in a dielectric medium that passes through: (**a**) an unladen conductive sphere and (**b**) a dielectric sphere with a permittivity lower than that of the environment.

The dielcometric method for monitoring the moisture content, component composition and other basic parameters of agricultural products has been of interest to researchers not only because of the potential to achieve a theoretically high speed, but because their working frequency is in the range 104–109 Hz and because of the low instrumental measurement error of 0.1% [73,74].

## 4. Discussion

There is a constant development of new technologies that are applied to the determination and measurement of moisture in grains. Despite the new alternatives and their characteristics, direct methods are still used in specialized laboratories and in the academic sector. Table 4 shows a comparison of the accuracy, operating range and response time of direct methods.

**Table 4.** Operational characteristics of direct methods for measuring moisture in grains.

| Method | Accuracy | Operating Range % | Response Time (Minutes) |
|---|---|---|---|
| Gravimetry/Oven | $0.09 \pm 0.16$ | 10–40 | 2280 |
| Distillation | $0.28 \pm 0.42$ | 10–35 | 60 |
| Coulometry Method (Karl Fischer) | $0.08 \pm 0.13$ | 10–35 | 45 |

The methods in Table 4 show a minimum measurement time of 20 min to obtain the measurement of grain moisture, which is a very accurate value. Due to this delay, it is not a method that can be easily used in continuous production lines, limiting its use in industry.

The accuracies in all cases are less than 0.3 percentage points in the percentage of humidity measured, which allows them to be used reliably and these methods can be used as a reference for other methods under study.

It is relevant to note that the measurements obtained by these methods may be different when the same sample is measured with a specific moisture content. To correct for differences in measurements, some authors suggest the use of homogeneous standards in the process relevant humidity determination range, and the subsequent application of linear regression that generates a specific equation that can be used to homologate all values in relation to the methodology used as a reference.

Indirect methods have a much faster response than previous methods, as shown in Table 5, where response times are shorter than those of direct methods. In addition, measurement ranges have been extended and measurements have decreased in accuracy.

**Table 5.** Operational characteristics of indirect methods of measuring grain moisture.

| Methods | Accuracy | Operating Range % | Response Time (minutes) |
|---|---|---|---|
| Electrical resistance/Impedance | ±0.3 | 1–44 | 0.166 |
| Capacitance | ±0.5 | 1–40 | 0.016 |
| Dielectric constant change | | | |
| Microwave | ±0.5 | 0–70 | 0.016 |
| Magnetic measurement method | | | |
| Magnetic resonance | 0.01 | 15–30 | 2 |
| Ultra-wideband (UWB) radar method | 1.0–1.4 | 1–26 | $8.3333 \times 10^{-10}$ |
| Ultrasonic (Acoustic Method) | ±0.5 | 10–60 | - |
| Infrared thermography (IR) | ±0.2 | 20–70 | 0.066 |
| Infrared spectroscopy | 0.09 ± 0.30 | 10–25 | 20 |
| Hyperspectral imaging (HSI) | 0.8 | – | 15 |
| Equilibrium relative humidity (ERH) | 3.15–3.59 | 1–30 | 20 |

Table 6 shows different response times according to the principle (first column) and its measurement procedures. Operating ranges have been limited to measurements according to suitable ranges within the industry for grain application. Accuracy is also varied with a plethora of electrical operating principle methods that stand out for their short response times.

**Table 6.** Operational characteristics of new grain moisture measurement technologies.

| Method | Accuracy | Operating Range % | Approximate Response Time (minutes) |
|---|---|---|---|
| RF-based moisture content and Artificial Neural Network | 98% | 0–30 | 1–2 |
| Prediction using ANN and SVR modeling techniques | +0.92–2.09 | 0.40 | 5–10 |
| Grain moisture determination by complex permittivity and compression | – | – | – |
| Microwave attenuation at 10.5 GHz and humidity density | ±0.07–0.52 | 12–26 | 1–2 |
| Temperature and humidity control | ±2.67–3.35 | 10–35 | 1–2 |
| Online measurement by resistance sensor | ±0.5 | 10–35 | 5–10 |

**Table 6.** *Cont.*

| Method | Accuracy | Operating Range % | Approximate Response Time (minutes) |
|---|---|---|---|
| Indirect determination of moisture using biospeckle technique | – | – | 0.166 |
| Moisture content in the mechanical properties of grains | ±0.0482–0.1547 | 0–14 | - |
| Moisture content measurement with inspection card | ±1.00 | 10–24 | 30–45 |
| Evaluation of grain moisture content by dielcometric method | 0.1 | 10–50 | 0–1 |

## 5. Results

A wide variety of principles and technologies for the determination of grain moisture have been found, with a greater number of developments related to the use of electromagnetic waves and the analysis of electrical characteristics (Figure 2). Variants of indirect methods have been adapted for dynamic measurements in grains or for measuring moisture in their silo storage cover.

Direct methods have remained unchanged and continue to be used for their reliability and accuracy despite the high response times they provide in their measurements.

New technologies have been found using chemical reactions, the use of laser light, image analysis and the use of neural networks and mathematical models for moisture prediction using physical parameters of the environment that affect grains. Methods are also added that consider the behavior of the mechanical properties of the grains, which are influenced by the existing humidity. It is clear that new technologies use technological equipment with greater data processing characteristics, but their accuracy does not exceed that of direct methods.

## 6. Conclusions

In the last two decades multiple methods have been proposed that allow determination of the humidity in grains. Several of these developments use non-destructive techniques that can be implemented in industrial processes. some of these developments take the measurement with the grain in motion and during its storage. These developments do not improve the accuracy and reliability of direct methods; however, the low response time provides advantages in their industrial application.

The use of neural networks and mathematical models for the estimation and prediction of moisture content has been considered; mostly the methods employing principles of electrical operation based on the electrical characteristics of the grain. In some cases, wireless communication technologies have also been implemented to introduce these developments into processing chains.

The proposed new methods do not have a standardization by accredited entities and their use requires a greater number of trials for a greater number of grain varieties so that their studies are more relevant and of greater fidelity in their application for the industry.

There are large numbers of portable equipment on the market that operate with the use of indirect methods and new technologies that provide measurements with low accuracies that in practice are used, justifying their practicality in terms of response times, portability and ease of operation, however the accuracy must be estimated with repetitive tests and correction factors.

**Author Contributions:** Conceptualization, O.F.; methodology, K.S. and L.R.; validation, H.P.; investigation, O.F. and K.S.; resources, M.G.; writing—original draft preparation, O.F. and K.J.; writing—review and editing, F.S.; supervision, H.P.; project administration, H.P. and O.F. All authors have read and agreed to the published version of the manuscript.

**Funding:** This research was funded by Universidad de Las Américas UDLA-Ecuador and Escuela Superior Politécnica del Litoral ESPOL.

**Conflicts of Interest:** The authors declare no conflict of interest.

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
