# Peer review of "New Sensing Technologies for Grain Moisture"

_agriculture, doi:10.3390/agriculture12030386_

Round 1

Reviewer 1 Report

In your paper "New Sensing Technologies of grain Moisture" you focus on the conventional and advanced technologies for estimation and sense of moisture content in grains. The topic of this manuscript is within the scope of Agriculture and the overall structure of the paper is suitable. I recommend corrections to make the manuscript better.

The abstract is highly summary but does not cover all the contents of the manuscript. So, the abstract is not easy to follow for the reader and does to really encourage to read the rest of the manuscript. More importantly, Line17-18 “It was found that technologies based on visual methods provide better accuracy and response times compared to other measurement principles.” But, There is no comparison of the accuracy of the different methods in the manuscript, and extreme caution should be exercised in drawing such conclusions.

Line 33-35 Repeat?  There seems to be a lot of typographical errors (duplication or missing) in this article.  

It is appropriate to discuss the importance of grain moisture content and its determination method in the introduction. The summary of Figure 2 is excellent, but the development and classification of MC determination methods are not well described.

Is “2.1.1.Gravimetric Method/Oven” different from the description in the first line of Table 1? Note the consistency of terminology in the manuscript

The MRI method seems to be missing in determining grain water content and water distribution

Part 3 seems to be an introduction to some speed measurement or prediction method, “Advances in grain moisture detection” may easily lead readers to misunderstand the relationship between these methods and the methods in Part 2, especially the indirect methods.

Author Response

Response to Reviewer 1 Comments

Point 1. In your paper "New Sensing Technologies of grain Moisture" you focus on the conventional and advanced technologies for estimation and sense of moisture content in grains. The topic of this manuscript is within the scope of Agriculture and the overall structure of the paper is suitable. I recommend corrections to make the manuscript better.

Response 1. Multiple corrections have been made according to the comments made and some relevant points have been structured

Point 2. The abstract is highly summary but does not cover all the contents of the manuscript. So, the abstract is not easy to follow for the reader and does to really encourage to read the rest of the manuscript. More importantly, Line17-18 “It was found that technologies based on visual methods provide better accuracy and response times compared to other measurement principles.” But, There is no comparison of the accuracy of the different methods in the manuscript, and extreme caution should be exercised in drawing such conclusions.

Response 2. The Abstract has been structured considering new technologies, applications, application principles and highlights the implementation of wireless communication technologies to implement some of these methods in dynamic production chains.

Point 3.  Line 33-35 Repeat?  There seems to be a lot of typographical errors (duplication or missing) in this article. 

Response 3. Fixed repeated lines and elements

Point 4. It is appropriate to discuss the importance of grain moisture content and its determination method in the introduction. The summary of Figure 2 is excellent, but the development and classification of MC determination methods are not well described.

Response 4.  The importance of determining grain moisture with respect to quality, safety and durability is cited, as well as the role they play with moisture and effects according to Figure 1.

Point 5. Is “2.1.1.Gravimetric Method/Oven” different from the description in the first line of Table 1? Note the consistency of terminology in the manuscript

Response 5.  Terminology has been unified

Point 6. The MRI method seems to be missing in determining grain water content and water distribution

Response 6.  Section 2.2.6 deals with the Magnetic Resonance method for the determination of moisture in the grain.

Point 7. Part 3 seems to be an introduction to some speed measurement or prediction method, “Advances in grain moisture detection” may easily lead readers to misunderstand the relationship between these methods and the methods in Part 2, especially the indirect methods.

Response 7.  The title has been changed to: New techniques to determine moisture in grains

Reviewer 2 Report

The current manuscript is focused on various moisture estimation techniques, covering the existing protocols and advancement in this field. The idea behind this manuscript is worth appreciating however, major correction is required.

General comments:

The manuscript needs keen attention to improve the standard of English. At many places sentences have an abrupt beginning and ending, making it confusing for the readers.

Specific comments:

The title of the manuscript should be corrected: “technologies of grain moisture” should be changed to “technologies for grain moisture”

Place the figure (Fig.1) after it is cited in the text for the first time.

Line 33-35: Extra lines need to be removed.

Line 49-50: Grain moisture content …. (14.5–15%). Please provide relevant citation.

Figure 2: change “Acustic” to “Acoustic”

Line 81: is it Figure 1 or Figure 2?

Line 82-84:  Direct methods ….. of grain. Using some techniques ……. from the species. Merge the two lines.

Place the table (Table 1) only after it cited in the text, for the first time.

Line 94: “placed in a cooker”. The author should not use the term cooker as this sounds inappropriate. It can be simply written as “hot air oven” or "temperature controlled oven".

Equation 1 is missing the “=” sign.

Line 98: while expressing the equation parameters, the units must be provided. Please follow this suggestion for all other equations within the manuscript.

Line 98: ??? is the wet basis content… it should be “wet basis moisture content”. Author should use a consistent abbreviation for representing moisture content such as “MC” or “mc”.

Line 118: Abbreviation should be declared where the referred term is used for the first time in the text.

Line 123: express all the equation parameters.

Line 134: Placing “Infrared spectroscopy” under the category of direct method is a huge mistake at the author’s end. The author should understand that the moisture meter (with IR heater) which operates on the principles of gravimetric analysis could be categorized under direct method. However, moisture estimation using IR spectroscopy would not qualify as a direct method.

Line 325: Authors may consider citing some of the recent advancement in moisture estimation such as one in the following link: https://www.sciencedirect.com/science/article/pii/S0260877421004155

Line 349: Cite the image source. This suggestion should be followed where the authors have taken the images from others work.

Line 370: What is “TIC’s” in Table 3? Please define the term.

Table 3: Authors must provide the required citation.

Line 554: section numbering is wrong.

Sections like “Results” and “Discussion” in a review paper format is quite uncommon. As the reported work is not a research outcome from the authors end, they should rename these sections accordingly. Moreover, the section numberings are wrongly placed. 

Author Response

Response to Reviewer 2 Comments

General comments:

Point 1. The manuscript needs keen attention to improve the standard of English. At many places sentences have an abrupt beginning and ending, making it confusing for the readers.

Response 1.  Revised and improved English language and spelling errors

Specific comments:

Point 2. The title of the manuscript should be corrected: “technologies of grain moisture” should be changed to “technologies for grain moisture”

Response 2. The change has been made to the title: "New Sensing Technologies for grain Moisture"

Point 3. Place the figure (Fig.1) after it is cited in the text for the first time.

Response 3. Figure 1 is cited on line 40, then Figure 1 is displayed

Point 4. Line 33-35: Extra lines need to be removed.

Point 4. Repeated lines have been removed

Point 5. Line 49-50: Grain moisture content …. (14.5–15%). Please provide relevant citation.

Response 5. Reference 5, state that: Grain moisture content at which the grains begin to breathe. more intensively is called the critical moisture (14.5–15%)

Point 6. Figure 2: change “Acustic” to “Acoustic”

Response 6. Changed the word to "Acoustic"

Point 7. Line 81: is it Figure 1 or Figure 2?

Response 7. The figure is no longer mentioned in that part of the text has been corrected.

Point 8. Line 82-84:  Direct methods ….. of grain. Using some techniques ……. from the species. Merge the two lines.

Response 8. Improved text and merged lines.

Point 9. Place the table (Table 1) only after it cited in the text, for the first time..

Response 9. The table is cited before it is submitted

Point 10. Line 94: “placed in a cooker”. The author should not use the term cooker as this sounds inappropriate. It can be simply written as “hot air oven” or "temperature controlled oven".

Response 10. It has been changed to: "that is placed in a hot air oven"

Point 11. Equation 1 is missing the “=” sign.

Response 11. Added the respective sign

Point 12. Line 98: while expressing the equation parameters, the units must be provided. Please follow this suggestion for all other equations within the manuscript.

Response 12. The equations have been revised and the units incorporated

Point 13. Line 98: ??? is the wet basis content… it should be “wet basis moisture content”. Author should use a consistent abbreviation for representing moisture content such as “MC” or “mc”.

Response 13. The suggested abbreviation has been used

Point 14. Line 118: Abbreviation should be declared where the referred term is used for the first time in the text.

Response 14. This aspect has been verified and corrected

Point 15. Line 123: express all the equation parameters.

Response 15. All elements of the equations have been described

Point 16. Line 134: Placing “Infrared spectroscopy” under the category of direct method is a huge mistake at the author’s end. The author should understand that the moisture meter (with IR heater) which operates on the principles of gravimetric analysis could be categorized under direct method. However, moisture estimation using IR spectroscopy would not qualify as a direct method.

Response 16. Infrared spectroscopy has been considered as an indirect method

Point 17. Line 325: Authors may consider citing some of the recent advancement in moisture estimation such as one in the following link: https://www.sciencedirect.com/science/article/pii/S0260877421004155

Response 17. The quote has been included in section 2.2.11

Point 18. Line 349: Cite the image source. This suggestion should be followed where the authors have taken the images from others work.

Response 18. All images are owned and drawn with changes

Point 19. Line 370: What is “TIC’s” in Table 3? Please define the term.

Response 19. It has been clarified in the document that ICTs means: Information and Communications Technology

Point 20. Table 3: Authors must provide the required citation.

Response 20. Table 3 is self-authored

Point 21. Line 554: section numbering is wrong

Response 21. Se ha corregido la numeración

Point 22. Sections like “Results” and “Discussion” in a review paper format is quite uncommon. As the reported work is not a research outcome from the authors end, they should rename these sections accordingly. Moreover, the section numberings are wrongly placed

Response 22. The title Discussion has been placed because in that section the benefit of technologies will be compared, for example in the benefits of the time-accuracy ratio of grain moisture measurement.

Reviewer 3 Report

  1. Figure 1, please add citation to Fig.1. please noted that citation should be given for all Figures and Tables.
  2. Table 1 Should be revised. If Infrared is direct technique, then, microwave and capacitance should be direct technique too.
  3. Citation should be provided in Table 1 and 2, along for each measurement technique.

Author Response

Response to Reviewer 3 Comments

Point 1. Figure 1, please add citation to Fig.1. please noted that citation should be given for all Figures and Tables.

Response 1. The figures are of own elaboration as well as the tables.

Point 2. Table 1 Should be revised. If Infrared is direct technique, then, microwave and capacitance should be direct technique too.

Response 2. This aspect has been corrected and infrared spectroscopy has been included as an indirect method

Point 3. Citation should be provided in Table 1 and 2, along for each measurement technique.

Response 3. Se ha incluido las citas en las tablas 1, 2 y 3.

Round 2

Reviewer 1 Report

The authors have addressed all the comments. Thanks to the author's work, it is of great significance for the application of relevant research fields to summarize and compare various methods for sensing technologies of grain moisture. Suggestions to be modified:

  1. The order of citation is wrong, please check it carefully and modify it. Including sequence and annotation methods

The paper can be accepted after the above corrections.

Author Response

Response to Reviewer 1 Comments

Point 1: The order of citation is wrong, please check it carefully and modify it. Including sequence and annotation methods

Response 1:  The references, bibliography, citation sequence and annotation have been reviewed.

Reviewer 2 Report

The authors have addressed all the comments. However, some minor corrections are still needed with the citations.

  1. The sequence of citations in the text is not maintained. 
  2. There are citations in the "Reference" section which are missing in the main text.
  3. There are a few citations that are wrongly numbered in the reference section. For example citation "90" in the text is presented as "92" in the reference section. Moreover, the author's name in the citation should be cross-checked. The authors may follow google scholar citations for the same. 

The paper can be accepted after the above corrections.

Author Response

Response to Reviewer 1 Comments

Point 1: The sequence of citations in the text is not maintained. 

Response 1: The citation sequence has been revised, the annotation has been ordered and improved.

Point 2: There are citations in the "Reference" section which are missing in the main text.

Response 2: Revised and removed references that are not cited in the main text

Point 3: There are a few citations that are wrongly numbered in the reference section. For example citation "90" in the text is presented as "92" in the reference section. Moreover, the author's name in the citation should be cross-checked. The authors may follow google scholar citations for the same.

Response 3: It has been verified that the citations correspond to the references and references that are not cited have been eliminated, the annotation of the citations in the text has been improved

Reviewer 3 Report

all questions were answered.

Author Response

Response to Reviewer 1 Comments

Point 1. all questions were answered.

Response 1: Thank you very much for your review
